# HPC: Hierarchical Progressive Coding Framework for Volumetric Video

Zihan Zheng*
Shanghai Jiao Tong
University
Shanghai, China
1364406834@sjtu.edu.cn

Houqiang Zhong*
Shanghai Jiao Tong
University
Shanghai, China
zhonghouqiang@sjtu.edu.cn

Qiang Hu[†]
Shanghai Jiao Tong
University
Shanghai, China
qiang.hu@sjtu.edu.cn

Xiaoyun Zhang
Shanghai Jiao Tong
University
Shanghai, China
xiaoyun.zhang@sjtu.edu.cn

Li Song
Shanghai Jiao Tong
University
Shanghai, China
song_li@sjtu.edu.cn

Ya Zhang
Shanghai Jiao Tong
University
Shanghai, China
ya_zhang@sjtu.edu.cn

Yanfeng Wang
Shanghai Jiao Tong
University
Shanghai, China
wangyanfeng@sjtu.edu.cn

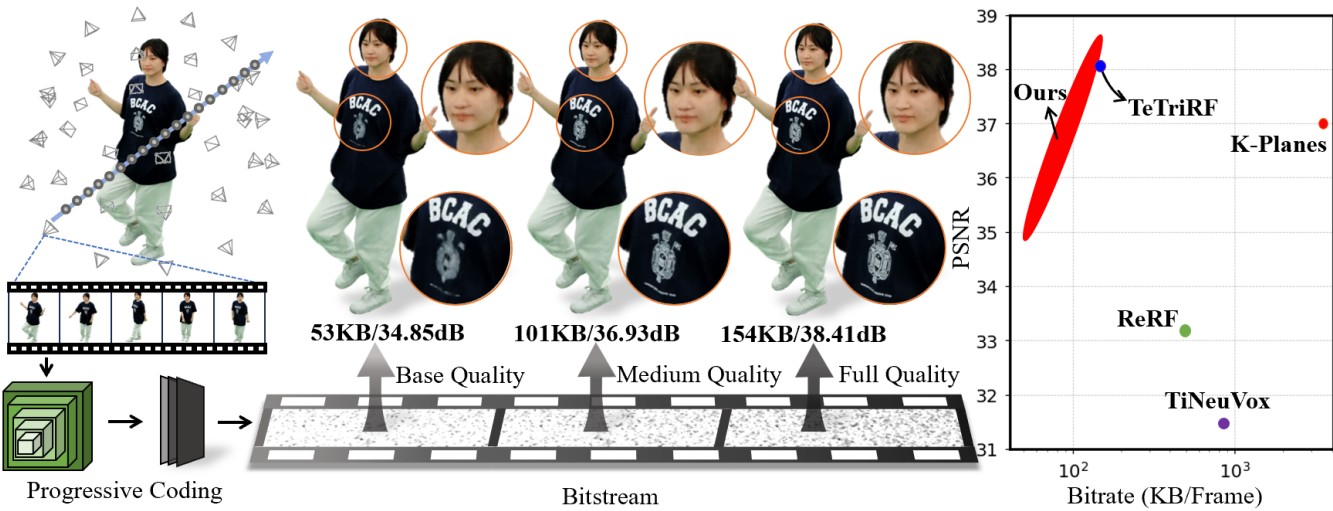

**Figure 1: Overview. With hierarchical representation (left), our HPC progressively encodes the volumetric video into a scalable bitrate bitstream using a single model, enabling different quality reconstructions (e.g., base quality 34.85dB@53KB, medium quality 36.93dB@101KB, full quality 38.41dB@154KB). The RD performance (right) still outperforms those fixed-bitrate methods (e.g., ReRF[65], TeTriRF[71]).**

## Abstract

Volumetric video based on Neural Radiance Field (NeRF) holds vast potential for various 3D applications, but its substantial data volume poses significant challenges for compression and transmission. Current NeRF compression lacks the flexibility to adjust video quality and bitrate within a single model for various network and device capacities. To address these issues, we propose HPC, a novel hierarchical progressive volumetric video coding framework achieving variable bitrate using a single model. Specifically, HPC introduces a hierarchical representation with a multi-resolution residual radiance field to reduce temporal redundancy in long-duration sequences while simultaneously generating various levels of detail. Then, we propose an end-to-end progressive learning approach with a multi-rate-distortion loss function to jointly optimize both hierarchical representation and compression. Our HPC trained only once can realize multiple compression levels, while the current methods need to train multiple fixed-bitrate models for different rate-distortion (RD) tradeoffs. Extensive experiments demonstrate that HPC achieves flexible quality levels with variable bitrate by a single model and exhibits competitive RD performance, even outperforming fixed-bitrate models across various datasets.

*Authors contributed equally to this work.
[†]Corresponding author.

*MM'24, October 28 - November 1, 2024, Melbourne, Australia.*
© 2024 Copyright held by the owner/author(s). Publication rights licensed to ACM.
ACM ISBN 979-8-4007-0686-8/24/10
https://doi.org/10.1145/3664647.3681107

## CCS Concepts

• **Information systems → Multimedia streaming**.

## Keywords

Volumetric Video, Dynamic NeRF, Progressive Coding, End-to-end Optimization

**ACM Reference Format:**
Zihan Zheng, Houqiang Zhong, Qiang Hu, Xiaoyun Zhang, Li Song, Ya Zhang, and Yanfeng Wang. 2024. HPC: Hierarchical Progressive Coding Framework for Volumetric Video. In *Proceedings of the 32nd ACM International Conference on Multimedia (MM '24), October 28-November 1, 2024, Melbourne, VIC, Australia.* ACM, New York, NY, USA, 10 pages. https://doi.org/10.1145/3664647.3681107

## 1  Introduction

Volumetric video captures dynamic 3D scenes, which allows users to freely select their viewing angles for a unique and immersive exploration experience. With its powerful 3D realism and interactive capabilities, volumetric video holds vast potential for 3D applications such as virtual reality, telepresence, sports broadcasting, remote teaching, and beyond. Therefore, volumetric video is considered a cornerstone for the next generation of media.

Recent advances in Neural Radiance Field (NeRF)[46] facilitate dynamic scene rendering for photorealistic volumetric video generation. Some methods[16, 37, 48, 52] utilize deformation fields to track voxel displacements relative to a canonical space, capturing motion information in volumetric video. However, their reliance on canonical space limits their effectiveness in sequences with large motion or topological changes. Other methods [16–18, 20, 22, 32, 70, 72] extend the radiance field to 4D spatio-temporal domains or introduce temporal voxel features, using a single neural network to fit 4D scenes and directly train on multi-view video data for high-quality temporal reconstruction. These methods effectively capture the dynamic details of scenes, but the substantial data volume poses significant challenges for transmitting volumetric video.

Several methods [15, 51, 55, 59, 65, 66, 71] have been developed to compress explicit features of dynamic NeRF for efficiently storing and transmitting volumetric video. ReRF[65] employs a compact motion grid and residual grid for representation, followed by traditional image encoding techniques to further reduce redundancy. TeTriRF[71] utilizes a three-plane decomposition of the representation and traditional video encoding methods, yielding improved results. However, they rely on traditional image/video encoding techniques and fail to jointly optimize the representation and compression of the radiance field, resulting in the loss of dynamic details and reduced compression efficiency. Additionally, they lack the flexibility to adjust video quality and bitrate within a single model for various network and device capacities. For achieving different bitrates, they require re-training and storing each model separately, resulting in large storage cost.

In this paper, we propose HPC, a novel hierarchical progressive volumetric video coding approach achieving variable bitrate using a single model. Our HPC improves coding efficiency and enables progressive variable bitrate streaming of data by being able

to scale the quality to available bandwidth or desired level of detail (LOD), see Fig. 1. Our key idea is to use multi-resolution feature grids which can be truncated at any level to achieve adaptive bitrate and quality. We hence introduce a hierarchical representation with multi-resolution residual feature grids to fully utilize feature relevance between consecutive frames. The feature grids of the representation are then sequentially quantized and entropy encoded to further reduce redundancy.

Moreover, we present an end-to-end progressive training scheme to jointly optimize both the hierarchical representation and compression, significantly enhancing the rate-distortion (RD) performance. Specifically, considering the non-differentiability of quantization and entropy encoding during compression, we introduce a method for simulating quantization and estimating bitrate, thus enabling gradient back-propagation. Additionally, we employ a multi-rate-distortion loss function together with a step-by-step training strategy to optimize the entire scheme. Experimental results demonstrate that our HPC achieves variable bitrate by a single model with higher compression efficiency compared to the fixed-bitrate models.

In summary, our contributions are as follows:

- We propose HPC, the first approach to enable progressive volumetric video coding, streaming and decoding. Our HPC achieves flexible quality levels and variable bitrate within a single model, while maintaining competitive RD performance.
- We present an efficient and compact hierarchical representation, which represents volumetric video as a multi-resolution residual radiance field with low temporal redundancy for high efficiency progressive compression.
- We introduce an end-to-end progressive learning approach that jointly optimizes hierarchical representation and compression based on a multi-rate-distortion loss function to enhance RD performance at each layer and overall.

## 2  Related work

### 2.1  Neural Scene Representation

NeRF[46] achieves photorealistic synthesis of new viewpoints using an implicit representation. This powerful method has quickly gained attention and has been extensively applied in various domains including pose estimation[10, 34, 64, 82, 83], 3D generative modeling[8, 23, 24, 27, 45], and 3D reconstruction[14, 33, 38, 61, 68, 69, 75]. In tasks involving the synthesis of new viewpoints in static scenes, NeRF-based approaches have recorded significant achievements. Differentiable rendering[10, 34, 49, 64, 82, 83] has demonstrated strong robustness against inaccuracies in camera pose inputs. A range of techniques[4–6, 43, 69] aimed at scene modeling has notably enhanced the quality of NeRF volumetric rendering. Additionally, various dense 3D representations such as octrees[19, 53, 67], multiscale hash tables[47], tensors[9, 60, 77], and mesh assets[11, 21, 54, 56, 75] have been explored to accelerate training and rendering. The breakthroughs achieved by NeRF in static scenes have spurred research into its application in dynamic scenes, laying a solid foundation for further advancements.

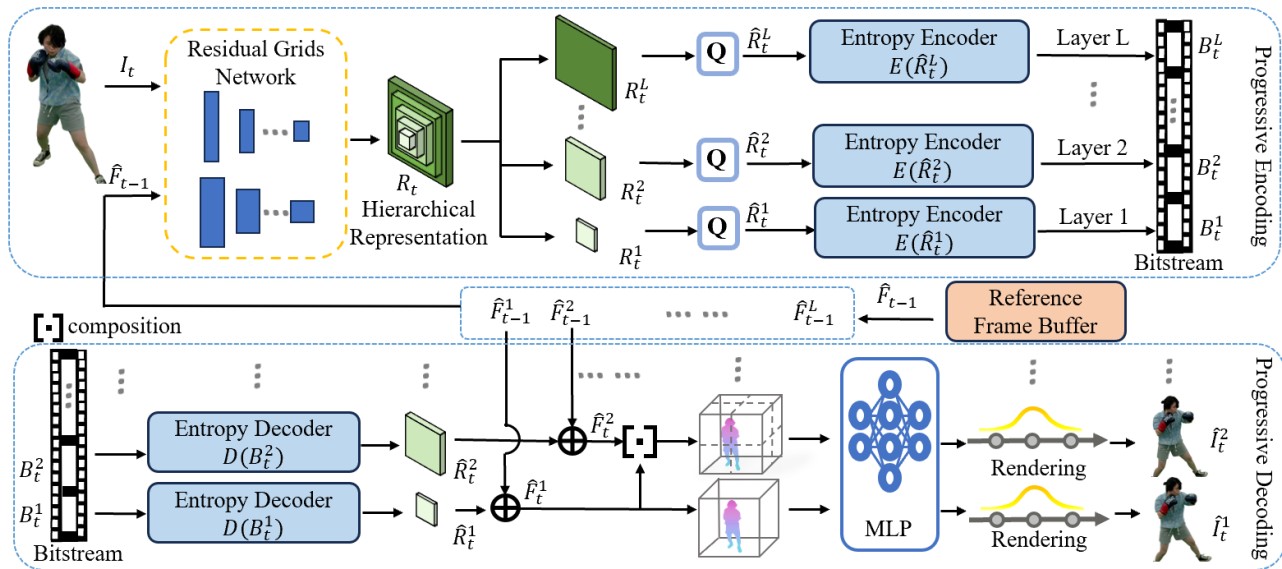

**Figure 2: Illustration of our HPC framework. In progressive encoding, residual grids network takes images $I_t$ and previous reconstructed feature grids $\hat{F}_{t-1}$ as input, generates multi-resolution residuals $R_t$. After quantization $Q$, the residuals are encoded into a bitstream $B_t$ via entropy encoder $E$. During progressive decoding, residuals are decoded from the bitstream and then recursively integrates with prior reference grids to recover the current frame features layer by layer.**

## 2.2 Dynamic Radiance Field Representation

Dynamic scenes present significant challenges, particularly when dealing with large motions on new view synthesis. Current advances in Neural Radiance Field (NeRF)[46] promote dynamic scene rendering for photorealistic volumetric video generation. The deformation field methods[16, 32, 48, 52, 59] recover temporal features by warping the live-frame space back into the canonical space, yet struggle with large motions and changes, leading to slower training and rendering. Another category of methods[7, 17, 18, 22, 30, 51, 53, 57, 70, 76, 78] extends the radiance field to 4D spatial-temporal domains, where they model the time-varying radiance field in a higher-dimensional feature space for quicker training and rendering, though at the cost of increased storage needs. Several works[65, 66, 71] adopt the residual radiance field technique by leveraging compact motion grids and residual feature grids to exploit inter-frame feature similarity, achieving favorable outcomes in representing long sequences of dynamic scenes. Our hierarchical representation further integrates the concept of residuals to correct errors and incorporate newly encountered regions, significantly reducing data redundancy. Moreover, this hierarchical representation enables us to implement progressive encoding, enhancing our capability to deliver optimal viewing experiences across varied network conditions and device capabilities.

## 2.3 NeRF Compression

In recent years, deep learning-based image and video compression methods[1–3, 12, 25, 26, 31, 35, 36, 39–41, 44, 58, 62, 63, 73, 74, 80] have been widely applied, achieving good rate-distortion (RD) performance on 2D videos. Currently, there are some efforts[15, 28, 29, 51, 55] underway to apply compression techniques in the NeRF

domain. Among them, VQRF[29] employs an entropy encoder to compress the static radiance field model, while ECRF[28] maps radiance field features to the frequency domain before applying entropy encoding. Despite their efficacy, these methods are still restricted to to static scenes, and lack exploration in dynamic spaces. ReRF[65], VideoRF[66], and TeTriRF[71] focus on dynamic scene modeling, employing traditional image/video encoding techniques for enhanced feature compression and do not simultaneously optimize both the representation and compression of the radiance field, leading to a loss of dynamic details and lower compression efficiency. Similar to approach[79, 81], we have designed a deep learning-based compression method for feature grids which can be optimized with representation and compressed for dynamic scenes, achieving very good RD performance.

## 3 Method

Our framework, depicted in Fig. 2, is organized into two core segments: hierarchical progressive encoding and hierarchical progressive decoding. The input includes multi-view images $I_t$ along with former reference feature grids. These inputs are processed through a residual grids network, designed to produce residual feature grids at multiple resolutions. These residual features are passed through entropy encoders and transmitted to the decoding side as a bitstream. During the decoding phase, the bitstream is decoded by entropy decoders into multi-resolutions residuals. They are integrated with former reference features to render images at different scales according to various bitrates. This process allows for the adaptive rendering of volumetric video outputs $\hat{I}_t$, catering to different compression needs and ensuring the high-quality content across varying resolutions. Next, we introduce the details about

the proposed hierarchical progressive encoding in Sec.3.1, consisting of hierarchical representation and entropy encoder, followed by hierarchical progressive decoding in Sec.3.2.

## 3.1 Hierarchical Progressive Encoding

**Hierarchical representation**. Recall that the NeRF-based representations map the implicit neural feature to color and density with MLP $\Phi$, where the feature $\mathbf{F}$ is determined by sampling position$(x, y, z)$ and view direction $\mathbf{d}$.

$$(\mathbf{c}, \sigma) = \Phi(\mathbf{F}(x, y, z, \mathbf{d})) \quad (1)$$

For long-sequence dynamic volumetric videos, our aim is to precisely establish the radiance field for each frame. Transitioning from a static to a dynamic scene highlights a significant challenge in data explosion. The simplistic method of transmitting individual per-frame feature grids $\mathbf{F}_t$ for a dynamic scene overlooks the essential aspect of temporal coherence, leading to considerable data redundancy. To mitigate the inefficiency of transmitting the entire radiance field for each frame and capitalize on the sequence's continuity, we segment the video sequence into multiple groups of features (GoFs), with each group comprising $N$ frames. Within these groups, take the first group for example $\mathbf{G}_1 = \{\mathbf{F}_1, \mathbf{R}_2, \mathbf{R}_3 \cdots \mathbf{R}_N\}$, we establish frame-to-frame residuals based on forward references, thus harnessing the inherent temporal coherence and substantially reducing the data required for accurate scene representation. And the feature grid ofthe frame $t$ can be recursively reconstructed by combining the feature grid of the previous frame with the current frame's residual $\mathbf{R}_t$.

$$\mathbf{F}_t = \mathbf{F}_{t-1} + \mathbf{R}_t \quad (2)$$

We decompose a large feature grid $\mathbf{F}$ into $L$ different resolutions to meet the needs for progressive streaming, $\mathbf{F} = \{\mathbf{F}^l \mid l \in [1, L]\}$. With the increment of the index $l$, each level of the feature grid $\mathbf{F}^l$ increases in resolution and enhances the details captured in the scene. By merging these feature grids from the lowest to the highest resolution, the reconstructed scene can be presented at different levels of precision. This methodology allows for the use of a single model to output features at different bitrates, satisfying the volumetric video viewing experience under various bandwidth conditions. Our multi-resolution residual representation is shown in Fig. 3. Within the GoFs, each group starts with a $L$-levels full feature grids as reference and follows with $L$-levels residuals. For frame $t$, $\mathbf{F}_t$ is reconstructed by adding together the residuals and the features from the previous frame at their respective levels. Our hierarchical representation with multi-resolution residual radiance fields, effectively reducing temporal redundancy in extended sequences and offering scalable quality adjustments. This approach tailors the streaming experience to fluctuating network conditions or specific requirements for the level of detail, enhancing both the efficiency and flexibility of volumetric video delivery.

$$\mathbf{F}_t = \{\mathbf{F}^l_{t-1} + \mathbf{R}^l_t \mid l \in [1, L]\} \quad (3)$$

**Entropy Encoder.** The sparsity of radiance field residual features significantly enhances compression and transmission efficiency. We scale the residuals of each level by the quantization parameter and round it to uint8, thereby substantially reducing the volume

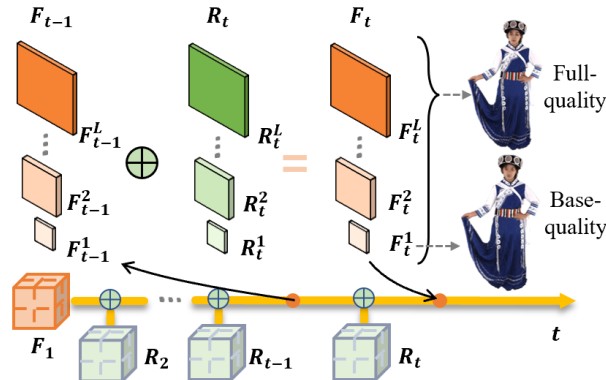

**Figure 3: The multi-layered feature grids for subsequent frames $\mathbf{F}_t$ can be recursively reconstructed by layer-wise accumulation of residuals $\mathbf{R}_t$.**

of data required for an accurate depiction. Following this quantization, the quantized residuals are subjected to compression via a range encoder[42], culminating in the generation of a more compact bitstream $B$. Notice that the residuals at each level are processed through separate entropy encoders rather than being amalgamated for collective compression. This layered approach ensures the unique statistical properties and predictability of the residuals at each resolution are meticulously accounted for, enabling more effective compression.

$$B_t = \{B_t^l \mid l \in [1, L]\} \quad (4)$$

$$B_t^l = \mathbf{E}^l \left( \mathrm{Q}(q \cdot \mathbf{R}_t^l) - \mathrm{Q}\left(q \cdot \min(\mathbf{R}_t^l)\right) \right) \quad (5)$$

where $\mathbf{E}^l$ is the $l$−th entropy encoder, and $\mathbf{Q}$ represents the quantization operation. In order to facilitate compression into the uint8 format, the data is first converted into non-negative values. The variable $q$ is the quantization parameter.During quantization, data is multiplied by $q$, effectively expanding the data range and enhancing quantization precision. By adjusting the parameter $q$, we trade-off between reconstruction quality and model storage.

## 3.2 Hierarchical Progressive Decoding

**Hierarchical Decoding.** On the decoding side, the user receives the transmitted bitstream and then decodes it using an entropy decoder to recover the original features. The operation is articulated as follows:

$$\hat{\mathbf{R}}_t = \frac{\mathrm{D}(B_t) + \mathrm{Q}(q \cdot \min(\mathbf{R}_t))}{q}, \quad (6)$$

where $\mathbf{D}$ is the entropy decoder. Since the data was converted into non-negative values and multiplied by $q$, we will revert it to its original range during decompression.

In practical scenarios, both the bandwidth available for data transmission and the computational power of decoding devices may be limited, insufficient to process all data. However, given that our scene representation is hierarchical, we can adaptively render dynamic scenes with varying effects at the decoding end, tailored to the device's computational capacity. Specifically, by selecting a smaller level $l$, we can choose to only receive and decode

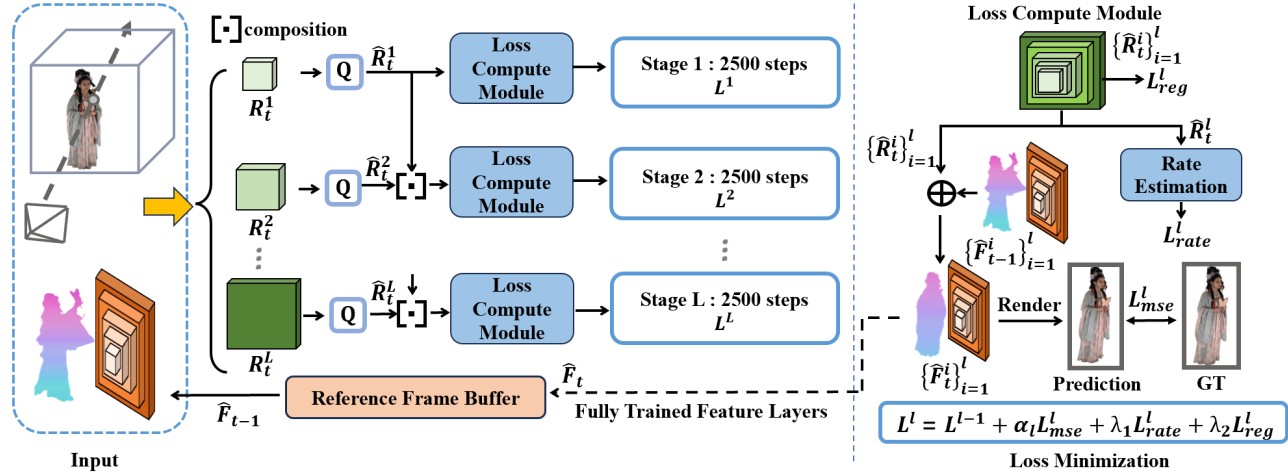

**Figure 4: Overview of our hierarchical progressive training. We generate different resolution feature grids $R_t^l$ from current frame images and previous reference feature $\hat{F}_{t-1}$ from buffer. The network trains on the most basic resolution grids, $l = 1$. As training advances, it progressively incorporates higher resolution grids from the next level, while supervising at each layer via the multi-rate-distortion loss $L^l$. After training is completed, the trained feature grids $\hat{F}_t^l$ are stored in the reference frame buffer.**

$\{\hat{F}^i \mid i \leq l\}$ and then render up to that selected resolution, ignoring the remaining higher-resolution $\{\hat{F}^i \mid i > l\}$. This approach facilitates a trade-off between rendering quality and model storage requirements.

We simultaneously transmit bitstream of varying resolutions, enabling the decoder to receive and process these data streams in parallel. Our methodology facilitates the implementation of LOD, progressively enhancing the granularity of scene details over time. This allows for a more efficient and dynamic presentation of complex scenes, aligning with the demands of high-fidelity visualization and real-time processing requirements.

**Rendering.** After decoding the reconstructed feature grids $\{\hat{F}^l\}$, we can obtain the corresponding color $c_i^l$ and density $\sigma_i^l$ through Eq.(1). Within a Group of Features (GoF), all frames and all levels of features are decoded and rendered through a global MLP $\Phi$. Then we proceed with volume rendering to obtain the rendering result. By accumulating the colors $c_i^l$ and densities $\sigma_i^l$ of all sampled points along a ray $\mathbf{r}$, we can derive the predicted color $\hat{c}^l(\mathbf{r})$ at resolution $l$ for the corresponding pixel:

$$\hat{c}^l(\mathbf{r}) = \sum_i^N T_i(1 - exp(-\sigma_i^l \delta_i))c_i^l, \tag{7}$$

where $T_i = exp(-\sum_{j=1}^{i-1} \sigma_j \delta_j)$, and $\delta_i$ denotes the distance between adjacent samples. In this way, we finally get the multi-resolution rendering results.

## 4 Hierarchical Progressive Training

In this section, we introduce our designed hierarchical progressive training methods, with the training process illustrated in Fig. 4. We train different feature grids progressively, jointly optimizing

reconstruction and compression. Our training approach is primarily divided into two parts: end-to-end joint optimization (Sec.4.1) and progressive training strategy (Sec.4.2).

### 4.1 End-to-end Joint Optimization

Here, we detail an end-to-end optimization strategy for enhancing compression efficiency by jointly optimizing the representation and compression of HPC. By applying simulated quantization to the feature grids $R_t$ and using an entropy model for bitrate estimation, we facilitate efficient end-to-end training. The objective of end-to-end joint optimization is to minimize the entropy of the radiance field representation while ensuring high reconstruction quality.

**Simulated Quantization.** Implementing quantization during the compression process significantly reduces the bitrate of feature grids at the expense of some information loss. By incorporating the quantization operation within the training phase, we enhance the model's robustness to the information loss caused by quantization. However, the rounding operation interrupts gradient flow, complicating end-to-end training. To avoid this, we emulate the effects of quantization by introducing uniform noise within the $[-\frac{1}{2}, \frac{1}{2}]$ range, as depicted in Eq.(8), allowing gradient flow to be preserved and supporting end-to-end training.

$$Q(x) = x + u, u \sim U(-\frac{1}{2}, \frac{1}{2}). \tag{8}$$

**Rate Estimation.** Entropy encoding on quantized feature grids produces a highly compressed bitstream. By incorporating bitrate measurement into the training phase and including it in the loss function, we encourage a distribution of features with lower entropy, effectively imposing a bitrate constraint during network updates. However, entropy encoding disrupts gradient flow, either. To address this, entropy models are introduced during training to

estimate the entropy of the grids, representing the compression bitrate's lower bound. These models are capable of approximating the probability mass function (PMF) for the quantized values $\hat{y}$ of a feature grid by calculating its cumulative distribution function (CDF), as demonstrated in Eq.(9). This approach enables network optimization towards lower bitrates while maintaining compatibility with gradient-based training.

$$P_{PMF}(\hat{y}) = P_{CDF}(\hat{y} + \frac{1}{2}) - P_{CDF}(\hat{y} - \frac{1}{2}). \tag{9}$$

To maintain precision, we refrain from the assumption of any predefined data distribution for the 3D grids. Instead, we construct a novel distribution within the entropy model to closely match the actual data distribution. The entropy model plays a crucial role during training by estimating the size of the compressed bitstream of $\mathbf{R}_t^l$ at resolution $l$, named $\mathcal{L}_{rate}^l$, which in turn informs the overall loss calculation.

$$\mathcal{L}_{rate}^l = -\frac{1}{L} \sum_{\hat{y} \in \hat{\mathbf{R}}_t^l} \log_2 \left( P_{PMF}(\hat{y}) \right) \tag{10}$$

### 4.2 Progressive Training Strategy

When end-to-end joint optimization is applied to feature grids across all resolutions within an entire scene, superior performance is achieved at the highest level. However, only supervising the overall reconstruction quality and simultaneously training all feature grids cannot ensure optimal results at every layer. In light of these challenges, we introduce our progressive training strategy, growing model with multi-level supervision.

Initially, the network trains only on the most basic resolution grids, $l = 1$. As training progresses, the model integrates higher resolution grids from the next $l + 1$ level, and supervises rendering for each layer of grids. Towards the final phases of training, we intentionally halt training on the low-resolution grids, and rate supervision is also progressively discontinued. This approach encourages the model to focus more on capturing finer details at the highest resolution grid.

**Training Objective.** The multi-rate-distortion loss function of each stage is defined as:

$$\mathcal{L}^l = \mathcal{L}^{l-1} + \alpha_l \mathcal{L}_{mse}^l + \lambda_1 \mathcal{L}_{rate}^l + \lambda_2 \mathcal{L}_{reg}^l, \tag{11}$$

$$\mathcal{L}_{mse}^l = \sum ||\mathbf{c}(\mathbf{r}) - \hat{\mathbf{c}}^l(\mathbf{r})||^2, \tag{12}$$

$$\mathcal{L}_{reg}^l = ||\hat{\mathbf{R}}_t^l||_1, \tag{13}$$

where $\mathcal{L}^l$ is the loss of level $l$, $\mathcal{L}_{mse}^l$ metrics the difference between the ground truth and the result of different levels of resolution rendered by our framework, measuring the quality of reconstruction. $\mathcal{L}_{rate}^l$ represents the estimated rate derived from $\hat{\mathbf{R}}_t^l$. $\mathcal{L}_{reg}^l$ is the L1 regularization applied to $\hat{\mathbf{R}}_t^l$ of different resolution to ensure temporal continuity and minimize the magnitude of $\hat{\mathbf{R}}_t^l$. The parameter $\lambda_1$ is used to balance the rate and distortion, allowing for control over the model size and reconstruction quality. The parameter $\lambda_2$ measures the extent of our constraint on $\hat{\mathbf{R}}_t^l$.

## 5 Experimental Results

### 5.1 Configurations

**Datasets.** In this section, we extensively assess our HPC framework on the ReRF[65] and DNA-Rendering[13] datasets. The ReRF dataset , $1920 \times 2080$, includes 74 camera views. We assign 70 for training and the remaining 4 for testing. The DNA-Rendering dataset, $2048 \times 2448$, includes 48 views, with 46 used for training and 2 for testing. For fairness, we specify the same bounding box for the same sequence in different comparison experiments.

**Setups.** In our framework, we specify the number of feature grids, $L$, as 6. Our experimental setup includes an Intel(R) Xeon(R) W-2245 CPU @ 3.90GHz and an RTX 3090 graphics card. During training, the initial settings are as follows: $\lambda_1$ and $\lambda_2$ are set to 0.000001, $\alpha_6$ (for full modeling) is set to 1, and the remaining $\alpha_l$ values are set at 0.25. The maximum number of iterations, $maxiter$, is set to $40,000$, with the activation iteration, $actiter$, at $2,500$. We utilize six distinct entropy models, each tailored to one of the six different-resolution feature grids. The duration for each GoF is consistently fixed at 20 frames.

### 5.2 Comparison

To our knowledge, HPC is the first framework for hierarchical progressive volumetric video coding, using a multi-resolution residual radiance field for optimized compression. It achieves variable RD performance through progressive encoding and decoding without additional training or compression. To validate HPC's effectiveness, we compare it at different resolutions with TiNeuVox[17], K-Planes[18], ReRF[65], and TeTriRF[71] both qualitatively and quantitatively. Figure 5 shows visual results of two sequences, demonstrating HPC's superiority in model compactness and detail precision.

**Table 1: Quantitative comparison against volumetric video encoding methods. We calculate the average PSNR, SSIM, and storage for each frame across all training and testing views separately.**

| Dataset | Method | Training View | | Testing View | | Size (MB)↓ |
|---------|--------|:---:|:---:|:---:|:---:|:---:|
| | | PSNR ↑ | SSIM ↑ | PSNR ↑ | SSIM ↑ | |
| ReRF | TiNeuVox[17] | 31.03 | 0.964 | 27.12 | 0.958 | 0.81 |
| | K-Planes[18] | 37.97 | 0.985 | 30.01 | 0.971 | 2.99 |
| | ReRF[65] | 33.04 | 0.979 | 30.88 | 0.975 | 0.50 |
| | TeTriRF[71] | 38.01 | 0.987 | 33.45 | 0.980 | **0.14** |
| | Ours | **38.38** | **0.990** | **33.55** | **0.981** | 0.14 |
| DNA-Rendering | TiNeuVox[17] | 29.28 | 0.953 | 22.19 | 0.947 | 0.80 |
| | K-Planes[18] | 31.98 | 0.979 | 27.81 | 0.962 | 3.00 |
| | ReRF[65] | 30.20 | 0.974 | 29.59 | 0.972 | 0.31 |
| | TeTriRF[71] | 32.33 | 0.980 | 29.48 | 0.973 | 0.16 |
| | Ours | **32.69** | **0.983** | **29.88** | **0.977** | **0.15** |

Besides qualitative experiments, we also conduct a quantitative comparison in terms of Peak Signal-to-Noise Ratio (**PSNR**), Structural Similarity Index (**SSIM**) and model storage as shown in Table 1. Our method shows significant advantages in reconstruction quality and model storage, achieving optimal rate-distortion performance. TiNeuVox[17] performs poorly on long sequences. K-Planes[18] has good reconstruction for known viewpoints but struggles with unknown ones and has a large model size. ReRF[65] has

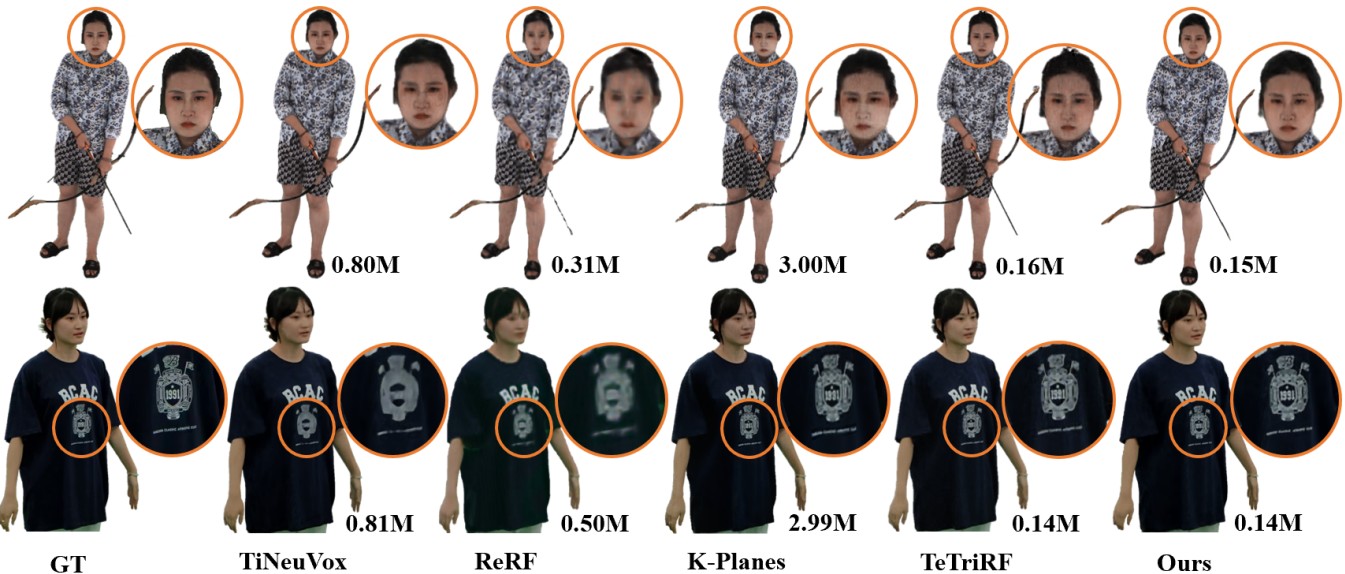

**Figure 5: Qualitative comparison against volumetric video coding methods TineuVox[17], K-Planes[18], ReRF[65], TeTriRF[71].**

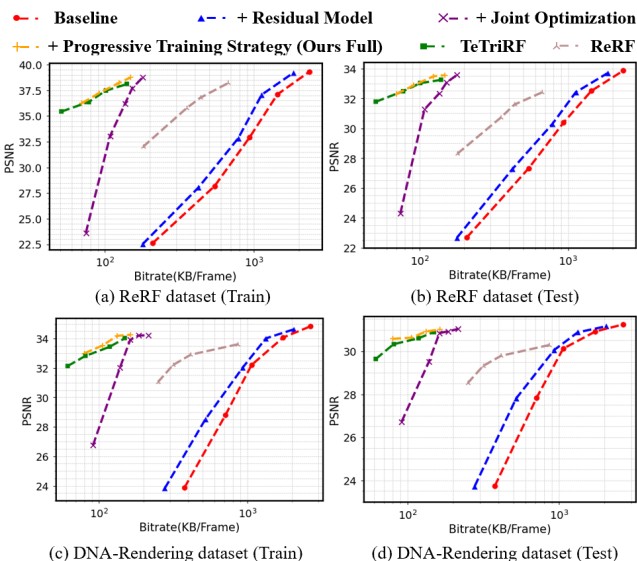

**Figure 6: Rate-distortion curves in both the ReRF and DNA-Rendering datasets. Rate-distortion curves not only illustrate the efficiency of various components within our method, but also demonstrate its superiority over ReRF[65] and TeTriRF[71].**

similar model storage to ours, but lower reconstruction quality. TeTriRF[71]'s RD performance is nearly comparable to ours but lacks progressive coding. Unlike ReRF[65] and TeTriRF[71], which require additional training and compression to adjust quality and storage trade-offs, our method optimizes this by selecting the number of feature grids during decoding.

**Table 2: The BDBR results of our HPC and TeTriRF[71] when compared with ReRF[65] on different datasets.**

| Dataset | Method | Training View | | Testing View | |
|---|---|---|---|---|---|
| | | BD-PSNR ↑ (dB) | BDBR ↓ (%) | BD-PSNR ↑ (dB) | BDBR ↓ (%) |
| ReRF | TeTriRF[71] | 6.582 | -80.822 | 4.878 | -86.948 |
| | Ours | **7.576** | **-81.523** | **5.173** | **-87.946** |
| DNA-Rendering | TeTriRF[71] | 5.516 | -80.575 | 4.037 | -88.682 |
| | Ours | **5.633** | **-82.463** | **4.398** | **-89.192** |

Fig. 6 and Table 2 also shows a comparison of RD performance between our HPC, ReRF[65] and TeTriRF[71]. We evaluate these methods using Bjontegaard Delta Bit-Rate (**BDBR**) and Bjontegaard Delta Peak Signal-to-Noise Ratio (**BD-PSNR**)[50]. From Table 2, we can see that on the ReRF dataset, we observe average BDBR reductions of 81.523% and 87.946% for training and testing views, respectively. Similarly, on the DNA-Rendering dataset, the average BDBR saving is 82.463% and 89.192% for training and testing views, respectively. Fig. 6 also demonstrates the superior RD performance of our method. Our method obviously performs better than ReRF[65] and have a slightly better result with TeTriRF[71]. However, all of them need to train multiple fixed-bitrate models for different rate-distortion tradeoffs. In contrast, Our method supports achieving multiple RD performances with a single compression and training process.

In additional, we evaluate the computational complexity of HPC at different quality levels, as detailed in the Table 3. Our decoding and rendering time gradually increases as reconstruction quality improves due to our use of higher-dimensional features to capture additional scene details, resulting in increased data volume and complexity required for decoding and rendering. Our method

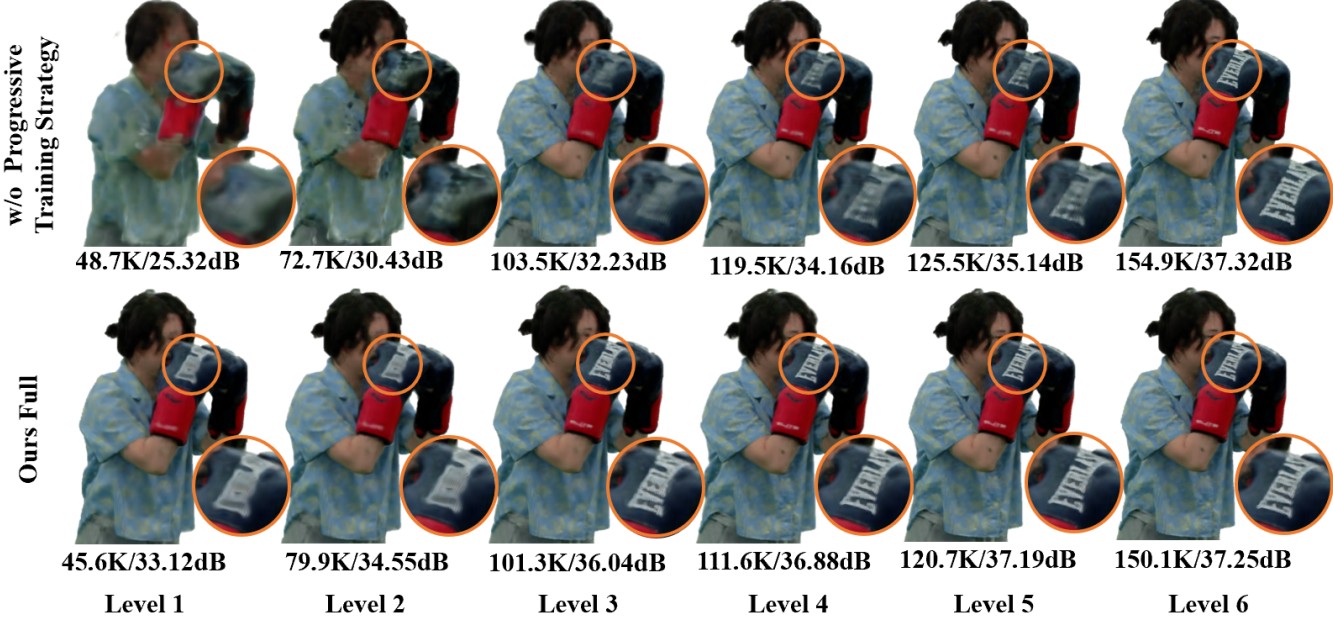

**Figure 7: The results of our full method in comparison to approaches without progressive training strategy. With the progressive training strategy, our HPC achieves flexible quality levels at variable bitrate using a single model.**

achieves shorter decoding and rendering times at base and medium reconstruction qualities, significantly outperforming TeTriRF[71] and remaining competitive with ReRF[65]. At full reconstruction quality, we achieve computational performance comparable to TeTriRF.

**Table 3: Computational complexity of our HPC at different quality levels in compression to ReRF[65] and TeTriRF[71].**

| Time | TeTriRF [71] | ReRF [65] | Ours | | |
|---|---|---|---|---|---|
| | | | Base Quality | Medium Quality | Full Quality |
| Decode(ms) | 101 | 55 | 22 | 43 | 121 |
| Render(ms) | 120 | 58 | 56 | 72 | 109 |

### 5.3 Ablation Studies

We conduct three ablation studies to validate the effectiveness of each component in our method. As a baseline, we opt for quantization and entropy encoding on the hierarchical representation and incrementally add each component under test to the baseline. Our primary focus lies on the dynamic residual model, end-to-end joint optimization, and progressive training strategy. In the first ablation study, we build on the baseline by adding the dynamic residual model. The second ablation study extends the first by incorporating an end-to-end joint optimization of the NeRF reconstruction and compression. Finally, in the third ablation study, which represents our full method, we further enhance the second study's setup by integrating the progressive training strategy.

The ablation study results are shown in Fig. 6. We adjusted the axis scales for clarity. Using the dynamic residual model, we represent non-keyframe features with small residual grids, significantly

reducing the overall model size. Joint optimization enhances rate-distortion performance and reduces storage while hardly affecting the reconstruction quality. Moreover, it offers advantages in terms of reconstruction quality over the baseline when we don't utilize all feature grids. Our progressive training strategy further decreases model size and improves robustness, enabling low-resolution grids to convey structural information effectively, whose importance can also be seen in Fig. 7 as HPC maintains excellent rendering quality across each level of result. Overall, the ablation studies highlight the critical roles of the dynamic residual model, joint optimization, and progressive training strategy.

### 6 Conclusion

In this paper, we propose HPC, the first progressive volumetric video coding approach, enabling flexible and effective scaling between quality and bitrate. HPC introduces a highly compact hierarchical representation with a multi-resolution residual radiance field to effectively leverage feature relevance between frames and generate different levels of detail. Furthermore, HPC employs an end-to-end progressive training scheme to jointly optimize the hierarchical representation and compression based on a multi-rate-distortion loss function, significantly improving RD performance of each level and overall. Experimental results show that HPC achieves variable bitrate using a single model and outperforms the state-of-the-art fixed-bitrate methods. HPC's unique variable bitrate capabilities enable progressive streaming and rendering across various quality levels, making it particularly suitable for scenarios with fluctuations in bandwidth and computational resources. This provides a fundamental basis for the widespread application of volumetric video.

## Acknowledgments

This work was supported by National Natural Science Foundation of China (62271308), STCSM (22511105700, 22DZ2229005), 111 plan (BP0719010), Open Project of National Key Laboratory of China (23Z670104657) and State Key Laboratory of UHD Video and Audio Production and Presentation.

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
