# OpenReview forum: "HPC: Hierarchical Progressive Coding Framework for Volumetric Video"
_acmmm.org/ACMMM/2024/Conference — MM2024 Oral_

### Official Review · Reviewer_6343 · 2024-05-22

**Rating:** 4
**Confidence:** 1

**Summary:**

The paper presents HPC, a novel Hierarchical Progressive Coding framework for volumetric video compression, leveraging a single model to adapt to varying bitrates and quality levels. HPC introduces a hierarchical representation with multi-resolution residual radiance fields, reducing temporal redundancy and enabling scalable quality adjustments. An end-to-end progressive learning approach with a multi-rate-distortion loss function jointly optimizes representation and compression. Extensive experiments demonstrate HPC's superior rate-distortion performance over state-of-the-art fixed-bitrate models like ReRF and TeTriRF, and ablation studies validate the effectiveness of its key components. While the approach adds complexity and increases training time, it offers significant flexibility and efficiency in handling different network conditions and device capacities.

**Strengths:**

1. Innovative Hierarchical Approach: The layer codec designed in the paper effectively reduces temporal redundancy and supports scalable quality adjustments.
2. Single Model Flexibility: Achieves multiple compression levels using a single trained model, offering significant flexibility and efficiency. While enabling multi-level compression in a single model, its performance is slightly better than TeTriRF.
3. Comprehensive Evaluation: Extensive experimental validation and ablation studies provide strong evidence of the method's effectiveness.

**Limitations:**

1. While the paper demonstrates superior rate-distortion performance and flexibility, it does not provide specific details on the decoding time or computational efficiency, which is important in 3D video streaming scenarios.
2. As an end-to-end codec, storing multi-resolution feature grids and entropy models might be too storage-consuming, which poses challenges in large-scale deployments.

**Suitability:**

3

---

### Official Review · Reviewer_PA36 · 2024-05-23

**Rating:** 5
**Confidence:** 2

**Summary:**

This paper introduces a hierarchical progressive volumetric video coding framework that enables flexible scaling between quality and bitrate. This framework utilizes a compact hierarchical representation with a multi-resolution residual radiance field, capitalizing on frame feature relevance to produce varying levels of detail.  Experimental results demonstrate the proposed method achieves variable bitrate with a single model, outperforming fixed-bitrate methods.

**Strengths:**

A. This paper proposes a hierarchical progressive volumetric video coding framework that achieves variable bitrate using a single model.

B. A hierarchical representation, incorporating a multi-resolution residual radiance field, is designed to minimize temporal redundancy in long-duration sequences.

C. An end-to-end progressive training scheme is introduced to jointly optimize the hierarchical representation and compression, leveraging a multi-rate-distortion loss function.

D. The performance of the proposed method surpasses recent benchmarks, demonstrating its effectiveness and superiority.

**Limitations:**

A、In this paper, the level L has been specifically designated as 6. However, it is crucial to justify the rationale behind this choice of level number. To ensure a comprehensive evaluation, the authors are recommended to consider alternative level settings, such as 7, and analyze their potential impact on the performance of the proposed method.
B、The Stage 1：0~2500 Steps and Stage 2：2500~5000Steps are shown in Figure 4, they should be " Stage 1：0~2500 Steps and Stage 2：2501~5000Steps" or "Stage 1：0~2499 Steps and Stage 2：2500~5000Steps". In addition, How is set for Stage l,  What is the detail allocation scheme?
C、The word Group of Features is repeatedly abbreviated. The format for the abbreviation in this paper should be consistent. (e.g., probability mass function (PMF), Group of Features (GoF)).

**Suitability:**

3

---

### Official Review · Reviewer_m27c · 2024-05-23

**Rating:** 4
**Confidence:** 3

**Summary:**

This paper proposed HPC, a hierarchical progressive coding framework for volumetric video. The proposed framework is designed for volumetric video based on Neural Radiance Field (NeRF). It introduced a hierarchical representation with a multi-resolution residual radiance field to generate different levels of detail, and to optimize the hierarchical representation and compression, an end-to-end progressive training scheme was designed. The proposed framework was compared with four related compression methods, TriNeuVox, K-Planes, ReRF, and TeTriRF. Evaluation results on two separate datasets have shown that HPC can achieve better compression results than TriNeuVox, K-Planes, and ReRF, and its RD performance is comparable with that of TeTriRF, which is a highly effective approach.

**Strengths:**

The major advantage of the proposed framework is that it can achieve variable bitrate encoding using a single model. This addresses a shortcoming of ReRF and TeTriRF which lack the capability for progressive coding.

The design of the architecture of HPC in Fig. 2 is reasonable. A major contribution of the framework is the hierarchical representation of dynamic scenes, and a progressive training strategy was also introduced and described in detail.

The paper also presented a comprehensive review of related studies in several domains, including neural scene representation, dynamic radiance field representation, and NeRF compression. It provides a good list of references for researchers working on related problems.

**Limitations:**

The paper lacks details in the entropy encoder part. It is not clear how the residuals were quantized into uint8. An equation explaining the quantization process would help the readers to understand it better. It is not clear what type of entropy encoder was implemented (e.g., Huffman coding, arithmetic coding, or something else). Also, for a fair comparison, it is worthwhile to mention the quantization and entropy coding strategies used in related methods such as ReRF and TeTriRF.

For the evaluation metrics in Section 5.2, the authors gave full names of PSNR and SSIM, but the metrics of BD-PSNR and BDBR are missing details.

Results on different quality levels and bitrates are only given for a single frame in Figure 7. The authors should give more quantitative results for the flexible quality levels at variable bitrates on all the 6 videos used for testing.

It would also be helpful to discuss the computational complexity of HPC compared to related methods like ReRF and TeTriRF.

**Suitability:**

3

---

### Official Review · Reviewer_T6Bw · 2024-05-27

**Rating:** 3
**Confidence:** 4

**Summary:**

This paper presents a progressive coding method for volumetric videos. The key ideas include the usage of hierarchical multi-resolution features and an effective end-to-end training strategy.

**Strengths:**

As claimed in the paper, this is one of the few works (the first work) that addresses the compression of volumetric rendering for multiview videos.
The design of the method is reasonable, and the results demonstrate its effectiveness.

**Limitations:**

The term “progressive coding” typically refers to a method where the quality of the reconstructed signal improves gradually as more data is received. However, in the context you’ve described, it seems more aligned with multi-resolution coding, where different levels of detail are transmitted

The proposed method appears to transmit multiple bitstreams of various resolutions, allowing flexible selection during decoding. While this approach benefits rendering, it does introduce additional computational tasks related to bitstream transmission, encoding, and decoding.

If the decoder only requires the smallest feature scale at time step t, how can it decode with two feature scales at t+1 without using the previous F_t (which contains both scales)? Did I miss something here?

If the method claims variable bitrate capability, it should include a mechanism to specify the expected bitrate. Without this, it becomes challenging to optimize the trade-off between quality and transmission efficiency. This discrepancy should be clarified in the paper to ensure a consistent understanding of the method.

Unfortunately, the paper doesn’t provide information on the timing complexity of the proposed method. It would be helpful if the authors addressed this.

The paper doesn’t explicitly compare the proposed method’s performance with neural multiview video codecs. It might be worth exploring this comparison in future research.

**Suitability:**

3

---

### Meta-Review · Area_Chair_mvqe · 2024-07-08

**Recommendation:** Accept (Oral)
**Confidence:** 5

**Metareview:**

The initial set of reviews was already more toward acceptance and the author's rebuttal helped to further clarify open issues leading to changes even more toward acceptance. Hence, the paper can be accepted.